# Spatial-Temporal Characteristics and Driving Factors of Coupling Coordination between the Digital Economy and Low-Carbon Development in the Yellow River Basin

**Zhenhua Xu and Fuyi Ci ***

College of Economics, Shandong Normal University, Jinan 250358, China
* Correspondence: cifuyi666@163.com; Tel.: +86-1785-4168-783

**Abstract:** Enhancing the level of coupling coordination between the digital economy and low-carbon development is not only an inevitable choice for implementing the strategy of ecological protection and high-quality development in the Yellow River Basin, but also a key path to achieve China's "Double Carbon" goal. The level of coupling coordination between the digital economy and low-carbon development in 78 cities in the Yellow River Basin from 2011 to 2020 is measured by a coupling coordination model, and the spatial-temporal characteristics and driving factors are analysed using the Dagum Gini coefficient, spatial autocorrelation model and geographic detector. This study found the following: (1) Rapid growth of the digital economy, with the slow growth of low-carbon development. The degree of coupling coordination of the two systems steadily improved and moved from a stage of near-disorder to primary coordination. (2) The degree of coupling coordination is spatially characterised by lower reaches > middle reaches > upper reaches, and provincial capitals and some coastal cities have a higher level of coupling coordination. Spatial differences in coupling coordination tend to widen, with inter-regional differences being the main source of overall differences. (3) There was a significant positive spatial correlation in the degree of coupling coordination. Local spatial clustering characteristics were dominated by High-High (H-H) clustering areas in Shandong and Low-Low (L-L) clustering areas in south-eastern Gansu. (4) The degree of coupling coordination was driven by both internal and external factors of the two systems, with internet penetration and the size of the telecommunications industry within the digital economy system as the most important factors driving the coupling coordination, and the interactions between the different drivers were all enhanced.

**Keywords:** Yellow River Basin; digital economy; low-carbon development; spatial-temporal characteristics; coupling coordination

## 1. Introduction

China has gradually entered the era of the digital economy, and also ushered in a new era of low carbon led by the "Double Carbon" goal, and digital economy construction and low-carbon development have become key tasks for economic and social transformation. Since China's official access to the international Internet in 1994, the digital transformation has continued to accelerate, with China's digital economy reaching CNY 45.5 trillion by 2021, accounting for 39.8% of its GDP, and the construction of a digital economy has achieved significant effects [1]. Meanwhile, China's low-carbon transition has achieved positive results in recent years, with China's carbon emission intensity falling by 26.4% in 2021 compared to 2012. However, total carbon emissions are still on an upward trend [2]. As a new engine for regional development [3], the digital economy, with its platform, speed and sharing characteristics, can not only effectively stimulate market economic vitality and social creativity [4], but also improve the efficiency of resource allocation and reduce energy consumption and carbon emissions in traditional industries. Thus, the digital economy is a boost to promote China's "Double Carbon" goal and achieve a zero-carbon economy [5]. Low-carbon development, as

the main development model of China's future economy and society, will force the low-carbon transformation of digital infrastructure, accelerate the process of industry digitisation and further promote the scale of digital industries [6]. This shows that there is an interactive coupling between the digital economy and low-carbon development.

The Yellow River Basin is an important economic development belt and an ecological security barrier for China [7]. With the digital economy accounting for more than 30% of GDP in 2021, it has become an important facilitator of economic development across the basin [8]. The Yellow River Basin hosts a concentration of China's coal industry, and the coal chemical industry accounts for more than 70% of the total carbon emissions nationwide, making a low-carbon transition urgent [9]. To achieve the goal of ecological protection and high-quality development in the Yellow River Basin, it is necessary to implement a low-carbon consensus in the construction of the digital economy, as well as to adhere to the digital drive in low-carbon development.

So, what is the coupling between the digital economy and low-carbon development? What is the degree of coupling coordination between the digital economy and low-carbon development in the Yellow River Basin? Are there regional differences and spatial effects? What are the main driving factors? To answer the above questions, the structure of this paper is presented as follows. The first part is an introduction, which briefly introduces the reasons for choosing the topic and the significance of the research. The second part is a literature review, which provides a comprehensive overview of scholars' research results on the digital economy and low-carbon development, and presents the innovation points of this paper. The third part is the coupling relationship, which analyses the interaction between the digital economy and low-carbon development. The fourth part is the research design, showing the regional overview, indicator system, research hypotheses, research methodology and data sources. Section 5 is the research results, analysing the time-series changes, regional differences, spatial effects and driving factors of the coupling coordination between the digital economy and low-carbon development in the Yellow River Basin. Section 6 is the conclusions and recommendations, which clarify the findings of this research and provides targeted recommendations. In addition, reflections on the limitations and directions for improvement of this paper are provided. The aim is to provide a scientific basis for enhancing the coordinated development of the Yellow River Basin in terms of digitalisation and decarbonisation.

## 2. Literature Review

Researchers have conducted extensive research on the definition [10], measurement methods [11] and the impact on the quality of economic development [12] of the digital economy. Robust results have been achieved in the areas of theoretical connotations [13], indicator systems [14] and influencing factors [15] of low-carbon development. However, research on the relationship between the digital economy and low-carbon development is still in its infancy and currently focuses broadly on three areas.

(1) The impact of the digital economy on low-carbon development. Academics generally believe that the digital economy is an important engine to promote the low-carbon transformation of the economy and society [16] and that the development of the digital economy can not only reduce the total amount of carbon emissions [17], but also reduce the intensity of carbon emissions [18], improve the efficiency of carbon emissions [19] and drive the development of low-carbon industries [2]. Other scholars argue that in the early stage of the development of the digital economy, the application and diffusion of digital technologies will enhance the production efficiency of highly polluting industries [20], leading to a sharp increase in energy consumption and carbon emissions [1]. When the digital economy reaches a certain level of development, the degree of digital industrialisation and digitisation of industry continues to deepen. The positive effects of capital and technology investment emerge, leading to increased energy efficiency and reduced carbon emissions, such that the impact of the digital economy on carbon emissions has an inverted "U" shape characteristic of "promoting increase first, then suppressing" [21].

Most scholars believe that the digital economy can achieve carbon emission reduction by promoting green technological progress [22], improving the energy structure [23], enhancing the efficiency of resource allocation [24] and promoting the upgrading of industrial structure [25]. Wang et al. (2022) found that the digital economy can deepen market integration, strengthen market competition, and enhance firms' innovation investment and innovation capacity, thereby improving their carbon reduction efficiency [26]. Wang et al. (2022) also analysed the impact of digital finance on household consumption carbon emissions from a household micro-perspective and found that digital finance contributed to the expansion of consumption as well as the upgrading of consumption propensity, which led to an increase in household consumption carbon emissions [27]. Chen et al. (2021) proposed the use of digital technologies, such as satellite remote sensing, to implement efficient and accurate dynamic monitoring of carbon sink resources that can improve carbon dioxide capture, utilisation and storage [28]. In addition, others have argued that there is industry heterogeneity and regional heterogeneity in the carbon reduction effects of the digital economy [29,30]. Furthermore, digital technologies can reduce the carbon intensity of neighbouring regions through spatial spillover effects [3].

(2) The impact of low-carbon development on the digital economy. Qu et al. (2022) argue that China's digital economy has not yet demonstrated green and low-carbon development qualities, and that energy consumption standards for digital products and digital infrastructure should be improved to promote the low-carbon development of China's digital economy [4]. Wang et al. (2022) propose that only by making the dual carbon goal a prerequisite for the healthy development of the digital economy can we create a good digital ecology and guarantee the accelerated evolution of China's new-generation digital information infrastructure towards a green and low-carbon direction [31].

(3) The interaction between the digital economy and low-carbon development. Although the synergy between the digital economy and the real economy [32], the green economy [33] and high-quality development [34] have been well discussed in academic circles, both theoretical and empirical studies have neglected the mission of "carbon emission reduction", and few studies have directly focused on the coupling coordination between the digital economy and low-carbon development. Zhang et al. (2014) elaborated the theory of symbiosis between smart cities and low-carbon cities from the perspective of information and communication and proposed that smart and low-carbon collaborative construction is an important direction for modern city development [35]. Zhang et al. (2014) analysed the double helix linkage development model between urban intelligence and low-carbon innovation, taking Tianjin city as the research object [36].

This paper differs from existing research in three main ways. First, most of the existing studies have explored the unidirectional impact of the digital economy on the low-carbon development subsystem (low-carbon economy, carbon emissions or carbon sinks), and the interaction between the two systems is only theoretically relevant, if at all. This paper provides a comprehensive account of the coupling relationship between the digital economy and low-carbon development from a systems theory perspective, and analyses the spatial and temporal characteristics and driving factors of the coupling coordination between the two systems through an empirical approach. Second, the indicator systems constructed in the existing literature for the digital economy and low-carbon development are still incomplete and hardly reflect the overall development level of the two systems comprehensively. Based on the connotation of the digital economy and low-carbon development, as well as the coupling relationship, this paper further improves the comprehensive evaluation index system of the two systems. Third, most of the existing studies on the digital economy or low-carbon development are focused at the national or provincial level. In this paper, we choose the Yellow River Basin as a special geo-economic area and conduct research based on city-scale data, which will not only help the cities in the Yellow River Basin to successfully implement their ecological protection and high-quality development strategies, but also accelerate the achievement of China's "Double Carbon" goal and narrow the gap between China's East, Middle and West.

## 3. Coupling Relationship

Coupling refers to the phenomenon of two or more systems interacting and influencing each other, and the coupling degree is usually used to indicate the strength of the interaction between the systems. The coupling coordination degree considers both the strength of the interaction between the systems and the level of their respective development, reflecting the degree of harmony and consistency between the systems in development [37]. The two systems, i.e., the digital economy and low-carbon development, influence each other. To reveal the coupling relationship, it is necessary to clarify how the two interact with each other and their linkage paths. The digital economy is a new economic form with data resources as the key element, modern information networks as the main carrier, and the integration of information and communication technology applications as an important driving force [38]. Existing studies summarise the digital economy into three aspects: digital infrastructure, industry digitisation and digital industrialisation [39]. Their inherent logic can be explained as follows: digital infrastructure is a prerequisite for the development of the digital economy—industry digitisation and digital industrialisation are the core of the development of the digital economy. Therefore, this paper divides the digital economy system into three parts: digital infrastructure, digitalisation of industry and digital industrialisation. Low-carbon development emphasises both low-carbon and development, and its essence lies in the transformation of production, living and ecological structures to improve carbon emission efficiency, enhance carbon sink capacity and promote sustainable economic and social development [40]. Therefore, this paper divides the low-carbon development system into three parts: low-carbon production, low-carbon living and low-carbon ecology.

### 3.1. Positive Effects of the Digital Economy on Low-Carbon Development

(1) The digital economy enhances low-carbon production. First, digital infrastructure provides a two-way flow of data for production, enhancing the exchange of information between the various production links. Online offices and video conferencing not only improve the overall level of synergy in the industry chain but also reduce carbon emissions in the intermediate links [41]. Second, with the support of digital technologies, such as cloud computing and artificial intelligence, the digital transformation of industries has been accelerated and the production systems of traditionally energy-intensive industries have approached optimisation, increasing energy use efficiency [42]. Third, digital finance accelerates the flow of funds across time and space, addresses problems of financing constraints and resource mismatch in the low-carbon sector, and enhances financial support for low-carbon innovation in new energy enterprises. Fourth, as the scale of the digital economy continues to expand, digital industries, such as telecommunications and information, are becoming pillar industries. The industrial structure is changing to a higher level, further reducing carbon emission intensity.

(2) The digital economy enhances low-carbon living. With the improvement of information networks and other communication facilities, new means of transactions, such as online shopping and mobile payments, are widely used and the trading of goods through online platforms can achieve precise matching between the supply side and the demand side of products. This not only helps to improve the quality of life but also reduces energy consumption and carbon emissions. Intelligent transportation systems can alleviate road congestion, improve the operational efficiency of the transportation system and achieve transportation carbon emission reduction [43]. The intelligent building process advances, and the establishment of a "smart building integrated management platform" to analyse and evaluate domestic energy data and provide effective domestic energy optimisation strategies will help improve the efficiency of domestic energy use.

(3) The digital economy enhances low-carbon ecology. Accelerating the capacity of ecological carbon sinks in forests, grasslands and oceans is key to achieving the goal of "Double Carbon", but due to the difficulty of obtaining ecological and environmental information, it is difficult to implement accurate monitoring of carbon sink resources.

Digital technology is used in the field of carbon sinks to address the problem of information asymmetry. Carbon footprint monitoring, carbon sink measurement and assessment with the help of a digital platform can effectively play a role in carbon sequestration in ecosystems and achieve objectives such as disaster prevention and control and ecological compensation [44].

### 3.2. Positive Effects of Low-Carbon Development on the Digital Economy

(1) Low-carbon development is forcing the transformation of digital infrastructure. Digital infrastructure is not only a driving force for low-carbon development but also a key area for energy saving and emission reduction, with digital equipment manufacturing and data centre operations generating significant energy consumption and carbon emissions. In the face of increasing pressure on carbon emissions, digital companies have committed to a low-carbon infrastructure transition by manufacturing efficient cooling equipment, creating energy monitoring systems and renewable energy projects, seeking to improve the efficiency of digital infrastructure energy use and achieving green, low-carbon development of digital infrastructure [45].

(2) Low-carbon development accelerates the digitalisation of industry. On the one hand, environmental regulation, pollution and carbon reduction and other related policies will force producers to take responsibility for reducing emissions and guide enterprises to use industrial internet, artificial intelligence and other digital technologies to low-carbonise traditional industries, thus accelerating the integration of digital technologies with traditional industries. On the other hand, as the public's awareness of low-carbon consumption grows, consumer demand for low-carbon products, such as new energy vehicles and energy-efficient home appliances, is increasing. This will lead digital technology to uncharted areas, accelerating the pace of innovation.

(3) Low-carbon development scales up digital industries. Low-carbon development has led to the large-scale application of information networks and digital technologies in economic, social and environmental areas. The universality of digital technologies has significantly reduced the investment costs and risks of digital industries, thus facilitating the expansion of digital industries such as telecommunications and software. Low-carbon development not only reduces total carbon emissions but also achieves sustainable economic and social benefits, so regions with high levels of low-carbon development can provide sufficient labour and financial support for the large-scale development of digital industries.

## 4. Research Design

### 4.1. Research Area

The Yellow River passes through nine provinces: Qinghai, Sichuan, Gansu, Ningxia, Inner Mongolia, Shaanxi, Shanxi, Henan and Shandong. Given that the vast majority of Sichuan Province is within the Yangtze River Basin, and that Hulunbeier, Chifeng and Tongliao cities in Inner Mongolia have been included in the north-eastern region (and given the availability of data), 78 prefecture-level cities in eight provinces (excluding Sichuan Province) were chosen. For related studies [46], 24 prefecture-level cities in Qinghai, Gansu, Ningxia and Inner Mongolia were classified as the upper reaches of the Yellow River Basin, 21 prefecture-level cities in Shaanxi and Shanxi were classified as the middle reaches of the Yellow River Basin, and 33 prefecture-level cities in Henan and Shandong were classified as the lower reaches of the Yellow River.

### 4.2. Indicator System

Based on the coupling relationship of the digital economy and low-carbon development, combined with relevant literature, a two-system comprehensive evaluation index system was constructed (Table 1).

**Table 1.** Comprehensive Evaluation Index System for Digital Economy and Low-carbon Development in the Yellow River Basin.

| Target Layer | Criterion Layer | Sub-Criterion Layer | Indicator Layer (Unit) | Attribute |
|---|---|---|---|---|
| Digital Economy system | Digital infrastructure | Mobile internet infrastructure | Mobile phone users per 10,000 (unit) | + |
| | | Broadband internet infrastructure | Internet broadband users per 10,000 (unit) | + |
| | Industrial digitalisation | The scale of rural e-commerce | Number of "Taobao Villages" (unit) | + |
| | | Enterprise digital transformation | Degree of digital technology application of listed companies (/) | + |
| | | Digital financial development | Total digital financial inclusion index (/) | + |
| | Digital industrialisation | Telecommunications industry development | Telecommunications business income (million CNY) | + |
| | | Information industry development | Number of employees in information transmission, computer service and software industries (10,000 people) | + |
| Low-carbon development system | Low-carbon production | Low-carbon production structure | Share of natural gas consumption (%) | + |
| | | | Tertiary industry output value as a proportion of GDP (%) | + |
| | | | The proportion of R&D input in fiscal expenditure (%) | + |
| | | Low-carbon production benefits | GDP per capita (CNY 10,000/person) | + |
| | | | Urban registered unemployment rate (%) | - |
| | | | Carbon emission intensity (tons/CNY 10,000) | - |
| | | | Energy intensity (tons/CNY 10,000) | - |
| | Low-carbon living | Low-carbon living structure | Gas penetration rate (%) | + |
| | | | Number of buses per 10,000 people(unit) | + |
| | | | Urban built-up land area per capita (sq.m/person) | + |
| | | Low-carbon living benefits | Urbanisation level (%) | + |
| | | | Engel coefficient (%) | - |
| | | | Income ratio between urban and rural residents (/) | - |
| | | | Carbon emissions per capita (tons/person) | - |
| | | | Energy consumption per capita (tons/person) | - |
| | Low-carbon ecology | Low-carbon ecological structure | Forest cover (%) | + |
| | | | Green area per capita (sq.m/person) | + |
| | | | The green coverage rate of built-up area (%) | + |
| | | Low-carbon ecological benefits | Industrial wastewater reuse rate (%) | + |
| | | | Harmless disposal rate of domestic waste (%) | + |
| | | | The integrated utilisation rate of industrial solid waste (%) | + |

Currently, there is no unified standard in academia for measuring the digital economy at the city level. Most scholars draw on Zhao et al. (2020) to select five indicators: the number of mobile phone users, the number of internet users, the number of people employed in the computer services and software industry, telecommunications business income and the digital inclusion index [47]. This paper adds two indicators, namely the number of "Taobao villages" and the degree of digital technology application of listed companies, and builds a comprehensive evaluation index system for the digital economy system from three dimensions: digital infrastructure, industry digitisation and digital industrialisation based on the connotations of the digital economy in the previous section. Among them, digital infrastructure mainly reflects the information network construction, with indicators selected from two aspects: mobile internet infrastructure and broadband internet infrastructure. Industry digitisation refers to the degree of application of digital technology in traditional industries such as agriculture, industry and services, with indicators selected from three aspects: the scale of rural e-commerce, digital transformation of enterprises and digital financial development. Digital industrialisation indicates the scale of development of digital industries such as telecommunications and information, with indicators selected from two aspects: the development of the telecommunications industry and the development of the information industry.

Some scholars have constructed a comprehensive evaluation index system for low-carbon development [48], but the selection and classification of indicators generally lack

scientific rigour. Based on the connotation of low-carbon development, indicators that are closely related to low-carbon as well as development have been selected. This paper divides low-carbon development into three subsystems: low-carbon production, low-carbon living and low-carbon ecology, and each subsystem has a comprehensive evaluation index system from two perspectives: structure and benefits. Among them, low-carbon production structure selects indicators closely related to energy saving and emission reduction from three aspects: energy structure, industrial structure and technology structure. The low-carbon production benefits section looks at economic growth, employment, carbon emissions and energy intensity. The low-carbon living structure has indicators selected from three main areas with high carbon emissions: food, housing and transport. Low-carbon living benefits focus on the level of urbanisation, the quality of life of the population, the urban–rural income gap and carbon emissions per capita. Low-carbon ecological structure examines green space and forest carbon sink cover. Considering that production and domestic waste, as an important part of ecosystems, can be recycled for artificial ecological management, thus reducing new energy development and carbon emissions, three indicators are selected for the low-carbon eco-efficiency section: reuse rate of industrial wastewater, the harmless disposal rate of domestic waste and the comprehensive utilisation rate of industrial solid waste.

### 4.3. Research Hypothesis

Based on existing research [49–53] and the coupling coordinated development of the two systems in the Yellow River Basin, as well as combining the weights of the previous indicators, a total of nine indicators were selected from both internal and external parts of the two systems. Among them, internet penetration rate X1 (number of Internet broadband subscribers per 10,000 people), development of digital finance X2 (digital inclusive finance index) and size of telecommunications industry X3 (telecommunications business revenue) were selected as internal drivers of the digital economic system. Upgrading of industrial structure X4 (proportion of tertiary industry output value), level of urbanisation X5 (proportion of the urban population to total urban and rural population) and ecological environment support X6 (green space per capita) were selected as internal drivers of the low-carbon development system. It is also necessary to enhance the macro-control ability of the government, as well as the resource allocation effect of the market and the active participation of the public. Therefore, the government regulatory capacity X7 (per capital fiscal expenditure), the degree of market development X8 (marketization index) and the level of human capital X9 (the number of university students per 10,000 people) were selected as the external drivers of the two systems [48].

#### 4.3.1. The Hypothesis of Internal Factors

1. Internet penetration rate. The internet is the basis for the development of the digital economy and an important carrier for realising the transmission of information elements [41]. The continued increase in internet penetration is conducive to accelerating the digital transformation of industries and expanding the scale of digital industrialisation. Meanwhile, the application of the internet has improved the efficiency of energy use and resource allocation, and has increasingly contributed to the low-carbon transformation of energy-intensive industries [22].

**Hypothesis 1 (H1).** *The higher the internet penetration rate, the stronger the contribution to the coupling coordination of the digital economy and low-carbon development.*

2. Development of digital finance. As an advanced form of finance, digital finance is an important driver for the development of the digital economy. Digital finance can provide the "financial blood" for digital infrastructure development and technology research, and can facilitate the low-carbon transformation of traditional industries and the innovative development of low-carbon industries by regulating the direction of funding [26].

**Hypothesis 2 (H2).** *The more developed the digital finance industry, the stronger the contribution to the coupling coordination of the digital economy and low-carbon development.*

3. Size of telecommunications industry. The telecommunications industry is the core industry of the digital economy, and the scale of the telecommunications industry directly determines the scale of the digital economy [17]. Meanwhile, the telecommunications industry has green and low-carbon attributes compared to traditional industries, and its carbon emission intensity is low [28]. Therefore, promoting the development of the telecommunications industry is a necessary path to realise the coupling coordination of the digital economy and low-carbon development.

**Hypothesis 3 (H3).** *The more developed the telecommunications industry, the stronger the contribution to the coupling coordination of the digital economy and low-carbon development.*

4. Upgrading of industrial structure. The industry is the carrier of digital economy construction and low-carbon development. Upgrading industrial structures can expand the application of digital technology and eliminate backward production capacity and improve energy use efficiency [54]. Therefore, upgrading the industrial structure will directly lead to a synergistic transformation of the digitalisation and low-carbonisation of industry.

**Hypothesis 4 (H4).** *The more advanced the industrial structure, the stronger the contribution to the coupling coordination of the digital economy and low-carbon development.*

5. Level of urbanisation. Regions with a high level of urbanisation have a greater demand for digital and low-carbon products from urban residents, and also provide sufficient labour for the construction of a digital economy and low-carbon development [55]. In addition, urbanisation is an important contributor to increased production efficiency and improved quality of life.

**Hypothesis 5 (H5).** *The higher the level of urbanisation, the stronger the contribution to the coupling coordination of the digital economy and low-carbon development.*

6. Ecological environment support. The process of ecological protection, restoration and construction needs strong support from digital technology, and offers a broad application prospect for digital technology [28]. Meanwhile, ecological protection is conducive to expanding environmental capacity and increasing carbon sink capacity [44]. The ecological environment is supportive of the synergistic convergence of the digital economy and low-carbon development.

**Hypothesis 6 (H6).** *The better the ecological environment, the stronger the contribution to the coupling coordination of the digital economy and low-carbon development.*

4.3.2. The Hypothesis of External Factors

1. Government regulatory capacity. Government is an important leader in the digital economy and low-carbon development [51]. The government can not only provide financial support for digital technology and low-carbon technology innovation, but also promote the digitalisation and low-carbon upgrading of traditional industries through rational planning and market supervision.

**Hypothesis 7 (H7).** *The stronger the government's regulatory capacity, the stronger the promotion of coupling the coordinated digital economy and low-carbon development.*

2. Degree of market development. The market is an important platform for enterprises to exchange and trade products. As marketisation accelerates, digital information elements

and resources are spreading and transferring among enterprises, thus accelerating the development of the digital economy. Meanwhile, under the constraint of the "Double Carbon" target, fierce market competition forces enterprises to strengthen their low-carbon innovation capabilities.

**Hypothesis 8 (H8).** *The higher the degree of marketisation, the stronger the contribution to the coupling coordination of the digital economy and low-carbon development.*

3. Level of human capital. High-quality talents are not only the foundation for the innovative development of the digital economy, but also the core force for leading the low-carbon transformation of the economy and society [56]. To promote the digital economy and low-carbon development, we need the joint support of "digital talents" and "low-carbon talents".

**Hypothesis 9 (H9).** *The higher the level of human capital, the stronger the contribution to the coupling coordination of the digital economy and low-carbon development.*

*4.4. Research Methodology*

4.4.1. Coupling Coordination Model

This study analyses the synergistic relationship between the digital economy and low-carbon development in 78 cities in the Yellow River Basin with the help of a coupling coordination model. The calculation steps are as follows.

1. Data standardisation. To maintain the consistency of various indicator data in terms of dimensions, the indicator data should be standardised. In the equation below, here $x_{ij}$ represents the initial value of the indicator; $X_{ij}$ represents the standard value of the indicator; $\min(x_{ij})$ represents the minimum value of the indicator; and $\max(x_{ij})$ represents the maximum value of the indicator. The formula is as follows:

$$X_{ij} = \frac{x_{ij} - \min(x_{ij})}{\max(x_{ij}) - \min(x_{ij})}, \ x_{ij} \text{ is a positive indicator} \tag{1}$$

$$X_{ij} = \frac{\max(x_{ij}) - x_{ij}}{\max(x_{ij}) - \min(x_{ij})}, \ x_{ij} \text{ is a negative indicator} \tag{2}$$

2. Entropy method. The weights of each indicator of the two systems of the digital economy and low-carbon development are calculated as $\lambda_{ai}$ and $\lambda_{bi}$, respectively, through the entropy weighting method [57], which results in a composite index $U_i$ for the digital economy and low-carbon development. The formula is as follows:

$$U_i = \sum_{j=1}^{n} \lambda_{ij} \cdot X_{ij} \tag{3}$$

3. Calculate the degree of coupling and the degree of coupling coordination. The formula is as follows:

$$C = 2 \times \left[ \frac{U_a U_b}{(U_a + U_b)^2} \right]^{1/2} \tag{4}$$

$$T = \alpha U_a + \beta U_b \tag{5}$$

$$D = (C \times T)^{1/2} \tag{6}$$

where $U_a$ and $U_b$ represent the digital economy index and low-carbon development index, respectively; $\alpha$ and $\beta$ represent the weighting coefficients of the two systems, respectively, taking $\alpha = \beta = 0.5$. C, T and D represent the coupling degree, comprehensive development index and coupling coordination degree, respectively. Since the coupling degree can only reflect the size of the interaction between the digital economy and low-carbon development,

and cannot indicate whether the two systems promote each other at a high value or suppress each other at a low value, the coupling coordination degree is selected to evaluate the level of coordination and interaction between the two systems. Concerning relevant studies [58], the coupling coordination was classified as follows: [0, 0.1) as extreme disorder, [0.1, 0.2) as severe disorder, [0.2, 0.3) as moderate disorder, [0.3, 0.4) as mild disorder, [0.4, 0.5) as near disorder, [0.5, 0.6) as basic coordination, [0.6, 0.7) as primary coordination, [0.7, 0.8) as intermediate coordination, [0.8, 0.9) as good coordination and [0.9, 1] as quality coordination.

### 4.4.2. Dagum Gini Coefficient

The Dagum Gini coefficient method was used to analyse the spatial differences and sources of differences in the coupling coordination of the digital economy and low-carbon development in the Yellow River Basin [34]. Here, overall differences = contribution of intra-regional differences + contribution of inter-regional differences + contribution of hypervariable density, i.e., $G = G_w + G_{nb} + G_l$. The formulas follow.

$$G = \frac{\sum_{i}^{h}\sum_{j}^{h}\sum_{l}^{n_i}\sum_{m}^{n_j}\left|y_{il} - y_{jm}\right|}{2n^2\overline{y}} \tag{7}$$

$$G_{ij} = \frac{\sum_{1}^{n_i}\sum_{m}^{n_j}\left|y_{il} - y_{jm}\right|}{n_i n_j\left(\overline{y}_i - \overline{y}_j\right)} \tag{8}$$

$$G_{ii} = \frac{\sum_{l=1}^{n_i}\sum_{m=1}^{n_i}\left|y_{il} - y_{im}\right|}{2n_i^2\overline{y}_i} \tag{9}$$

$$G_w = \sum_{i=1}^{h} G_{ii}p_i s_i \tag{10}$$

$$G_{nb} = \sum_{i=2}^{h}\sum_{j=1}^{i-1} G_{ij}\left(p_i s_j + p_j s_i\right)D_{ij} \tag{11}$$

$$G_l = \sum_{i=2}^{h}\sum_{j=1}^{i-1} G_{ij}\left(p_i s_j + p_j s_i\right)\left(1 - D_{ij}\right) \tag{12}$$

$$D_{ij} = \frac{d_{ij} - p_{ij}}{d_{ij} + p_{ij}} \tag{13}$$

$$d_{ij} = \int_0^\infty dF_i(y)\int_0^y (y - x)dF_j(x) \tag{14}$$

$$p_{ij} = \int_0^\infty dF_i(y)\int_0^y (y - x)dF_i(x) \tag{15}$$

In Equation (7), G is the overall Gini coefficient, h is the number of regions, n is the number of cities, i and j are the inner sets of region h, l and m are different cities, and $y_{il}$ and $y_{jm}$ are the coupling coordination of l city in i region and m city in j region, respectively. In Equations (8) and (9), $G_{ii}$ is the Gini coefficient of the i region, and $G_{ij}$ is the i and j inter-regional. In Equations (10)–(12), $G_w$, $G_{nb}$ and $G_l$ are the intra-regional difference contributions, inter-regional differences contribution and hypervariable density contribution, respectively. In Equation (14), $d_{ij}$ is the difference between i and j coupling

coordination degrees, i.e., the sample values of $y_{il} - y_{jm} > 0$ plus the total mathematical expectation. In Equation (15), $p_{ij}$ is the summed mathematical expectation of all sample values for $y_{jm} - y_{il} > 0$ in i and j regions, and $F_i$ is the cumulative density distribution function for i region.

### 4.4.3. Spatial Autocorrelation Model

1. Global spatial autocorrelation. The degree of coupling coordination between the digital economy and low-carbon development in the Yellow River Basin may be spatially autocorrelated. Therefore, the global spatial autocorrelation model is used to test the overall association of the coupling coordination degree, which is expressed by global Moran's I with the following formula:

$$I = \frac{\sum_{i=1}^{n}\sum_{j=1}^{n}W_{ij}(x_i - \overline{x})(x_j - \overline{x})}{S^2\sum_{i=1}^{n}\sum_{j=1}^{n}W_{ij}} \tag{16}$$

where $W_{ij}$ is the adjacency matrix; n is the number of specimens; x is the observed value; $\overline{x}$ is the mean; and $S^2$ is the sample variance. $I \in [-1, 1]$, $I > 0$ indicates a positive correlation in space, that is, the high-value region of the coupling coordination degree is adjacent to the high-value area, and the low-value area is adjacent to the low-value area; $I < 0$ indicates a negative spatial correlation, that is, the high-value area of coupling coordination is adjacent to the low-value area; $I = 0$ indicates a random distribution, that is, there is no spatial correlation in the degree of coupling coordination.

2. Local spatial autocorrelation. Since the global spatial autocorrelation can only reflect the overall association state of the coupling coordination degree, but cannot show the heterogeneity characteristics of each city, the global Moran's I is decomposed into each unit, and the local Moran's I of the coupling coordination degree of the city is calculated and analysed in combination with the LISA clustering map, and the formula is as follows:

$$I_i = \frac{x_i - \overline{x}}{S^2}\sum_{j=1}^{n}W_{ij}(x_j - \overline{x}) \tag{17}$$

where the meaning of each variable is consistent with the meaning of the variable in Equation (16). Among them, H (L)-H (L) clustering indicates that the city's coupling coordination degree is high (low), and the coupling coordination degree of neighbouring cities is high (low); H (L)-L (H) clustering indicates that the city's coupling coordination degree is high (low), and the adjacent city's coupling coordination degree is low (high).

### 4.4.4. Geodetectors

The factor detection and interaction detection parts in the geodetectors were used to detect the consistency of the spatial distribution pattern of each driving factor and the coupling coordination degree. Further, the influence of each driving factor on the coupling coordination degree of the digital economy and low-carbon development was analysed [59]. The formula is as follows:

$$q = 1 - \frac{1}{N\sigma^2}\sum_{h=1}^{L}N_h\sigma_h^2 \tag{18}$$

where q represents the driver influence coefficient; N and $N_h$ represent the total number of samples and the number of samples in the h region, respectively; L represents the number of subregions; $\sigma^2$ and $\sigma_h^2$ represent the variance of basin-wide coupling coordination and the variance of subregional coupling coordination, respectively; and q lies between 0 and 1, with q-value proportional to the driving force.

*4.5. Data Sources*

The data used in the article are mainly from the China City Statistical Yearbook, the China City Construction Statistical Yearbook and the statistical yearbooks of various cities. Among them, the degree of digital technology application of listed companies (by calculating the frequency of AI technology, blockchain technology, cloud computing technology, big data technology and digital technology application breakdowns in listed companies' reports, which are then aggregated on average to the city level), the digital financial inclusion index and the number of "Taobao villages" are obtained from the CSMAR database, the website of the Digital Finance Research Centre of Peking University and the research report of China Taobao villages. In addition, urban carbon emissions mainly come from direct energy (Gas and LPG) consumption and indirect energy (electricity) consumption. Referring to Yao et al. (2022), the various types of energy consumption in the city were multiplied by their corresponding carbon emission factors (carbon emission factors published by the IPCC in 2006) and summed to obtain the carbon emissions for the year [60]. Missing indicators for individual regions are supplemented by interpolation.

## 5. Research Results

*5.1. Spatial-Temporal Characteristics of Coupling Coordination*

Based on the comprehensive evaluation index system of the digital economy and low-carbon development, the degree of coupling coordination between the digital economy and low-carbon development in the Yellow River basin as a whole, as well as in the upper, middle and lower reaches from 2011 to 2020, was measured by Equations (1)–(6) (Table 2, Figure 1).

**Table 2.** Time Trends of Coupling Coordination in the Yellow River Basin.

| Year | $U_a$ | $U_b$ | $U_a/U_b$ | D | Type of Coordination |
|------|-------|-------|-----------|---|----------------------|
| 2011 | 0.1594 | 0.3359 | 0.4745 | 0.4768 | On the verge of disorder |
| 2012 | 0.1795 | 0.3464 | 0.5182 | 0.4948 | On the verge of disorder |
| 2013 | 0.2019 | 0.3575 | 0.5648 | 0.5134 | Basic coordination |
| 2014 | 0.2123 | 0.3683 | 0.5764 | 0.5237 | Basic coordination |
| 2015 | 0.2275 | 0.3760 | 0.6051 | 0.5354 | Basic coordination |
| 2016 | 0.2462 | 0.3860 | 0.6378 | 0.5492 | Basic coordination |
| 2017 | 0.2652 | 0.4003 | 0.6625 | 0.5646 | Basic coordination |
| 2018 | 0.2849 | 0.4022 | 0.7084 | 0.5748 | Basic coordination |
| 2019 | 0.3080 | 0.4120 | 0.7476 | 0.5890 | Basic coordination |
| 2020 | 0.3283 | 0.4248 | 0.7728 | 0.6027 | Primary coordination |

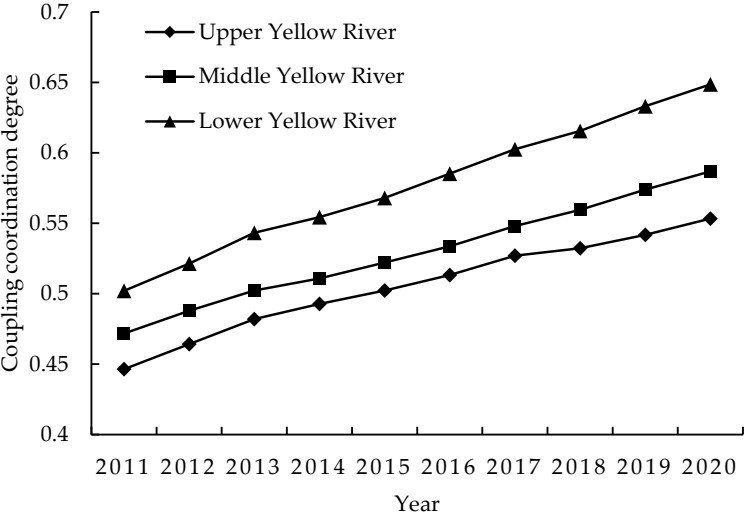

**Figure 1.** Trends of Coupling Coordination between the Upper, Middle and Lower Reaches in the Yellow River.

5.1.1. Time-Series Characteristics

As shown in Table 2, the overall digital economy index ($U_a$) and the low-carbon development index ($U_b$) both increased from 2011 to 2020. Among them, the digital economy index grew rapidly, from 0.1549 in 2011 to 0.3283 in 2020, with an average annual growth rate of 10.60%; the low-carbon development index grew slowly, from 0.3359 in 2011 to 0.4248 in 2020, an average annual increase of 2.65%. As the development of the digital economy started late, there was still a large gap between it and low-carbon development, but the difference between the two systems narrowed as the pace of digital economy development accelerates. The overall coupling coordination degree grows during the study period, increasing from 0.4768 in 2011 to 0.6027 in 2020, with a total increase of 26.41%. The type of coupling coordination transitions from near disorder to primary coordination, indicating that the driving force of the digital economy on low-carbon development and the leading power of low-carbon development to the digital economy increased.

In terms of stages, 2011–2012 was the stage of the near disorder, during which the construction of the digital economy was still in its infancy, while the energy consumption and carbon emission intensity of the traditional manufacturing industry remained high. The promotion effect of the digital economy on low-carbon development had not yet been fully released, making the level of coupling coordination between the two systems relatively low. The period 2013–2019 was a basic coordination phase, in which governments strongly supported the integration of digital technologies, such as the internet, with the real economy, which significantly improved the energy use efficiency of traditional industries. The digital economy and low-carbon development were relatively coordinated. In 2020, the primary coordination stage was reached, the development of China's digital economy, stimulated by the new epidemic, accelerated, and the digitalisation of industries deepened. The "Double Carbon" target accelerated the transformation of the digital infrastructure into low-carbon, and the coupling coordination of the digital economy and low-carbon development entered a new stage. In sum, the level of coupling coordination between the digital economy and low-carbon development in the Yellow River Basin improved, but the average level (mean value of 0.5424) remained low.

5.1.2. Spatial Divergence Characteristics

1. Three regional dimensions. As shown in Figure 1, the degree of coupling coordination between the digital economy and low-carbon development in the upper, middle and lower reaches of the Yellow River Basin increased year by year from 2011 to 2020. Among them, the upper reaches of the Yellow River rose from 0.4464 in 2011 to 0.5533 in 2020, an average annual increase of 2.39%; the middle reaches of the Yellow River increased from 0.4717 in 2011 to 0.5868 in 2020, an average annual increase of 2.44%; the lower reaches of the Yellow River increased from 0.5020 in 2011 to 0.6486 in 2020, an average annual increase of 2.92%. The mean values were 0.5055, 0.5297 and 0.5774 for the upper, middle and lower reaches, respectively. A reason for this is that Shandong and Henan provinces in the lower reaches of the Yellow River Basin have strong economies, sound infrastructure and sufficient human resources, and the digital transformation of industries and low-carbon process innovation is faster. The development of the digital economy in the middle reaches increased, but digital technology is still under-invested in the production sector. The construction of digital infrastructure consumes large amounts of energy, and the synergy between the digital economy and low-carbon development needs to be further enhanced. The upper reaches were constrained by factors such as location and transportation and lack of talent and technology—the process of building digital infrastructure is relatively slow. The digital economy is not sufficiently supportive of economic, social and environmental systems, which is not conducive to the improvement of the level of coupling coordination of the two systems.

2. Urban dimension. To further analyse the spatial evolution of the level of coupling coordination between the digital economy and low-carbon development in the 78 cities, a visual representation of the type of coupling coordination between each city in 2011, 2014,

2017 and 2020 was carried out using ArcGIS 10.2 software (Figure 2). In 2011, Qingdao, Jinan, Zhengzhou and Xi'an demonstrated the primary coordination type. Capital cities in the middle and upper reaches of the province, such as Taiyuan and Lanzhou, and coastal cities, such as Yantai, Weifang and Rizhao, were of the basic coordination type. Most of the remaining cities were of the near disorder type. From 2011 to 2014, Qingdao, Zhengzhou and Xi'an were the intermediate coordination type. Five cities, including Yantai, Taiyuan and Lanzhou, were the primary coordination type. The number of basic coordination-type cities increased from 14 in 2011 to 40, with the regional scope spreading from the eastern coastal region to the central inland region. From 2014 to 2017, Jinan became an intermediate coordination type. The number of cities with the primary coordination type increased to 13. The number of cities with the basic coordination type increased to 52, and the regional scope further extended to the western region. Only nine cities of the near-disorder type remained, mainly concentrated in south-eastern Gansu. From 2017 to 2020, Qingdao, Jinan, Zhengzhou and Xi'an became the good coordination type. Five cities, including Yantai, Taiyuan and Lanzhou, increased to the intermediate coordination type. The number of cities in the primary coordination type increased to 26, mainly concentrated in the lower reaches; most of the remaining cities become the basic coordination type. In summary, high values of coupling coordination were mainly concentrated in provincial capitals and some coastal cities, while most other cities had a low level of coupling coordination. Over time, the level of coupling coordination of all cities increased to a certain extent. By 2020 most cities had a coordinated development phase, but the coupled coordination in the strong economic areas grew more rapidly, resulting in a polarisation of the "strong get stronger, weak get weaker".

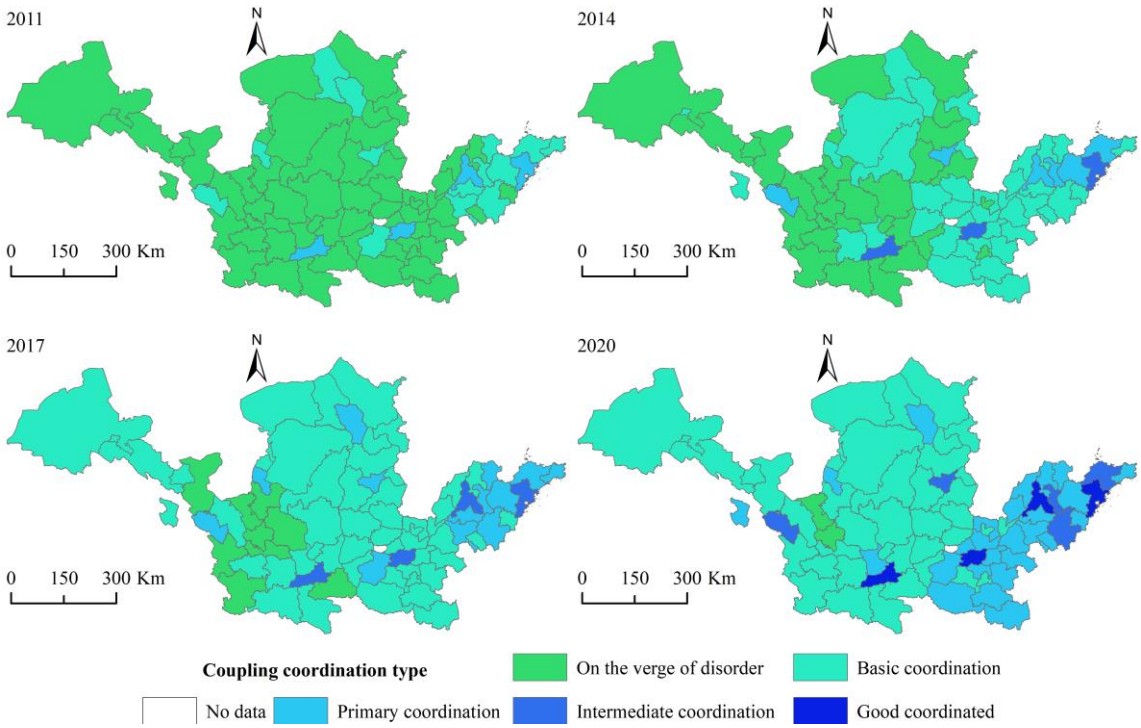

**Figure 2.** Spatial Characteristics of the Coupling Coordination in the Yellow River Basin.

5.1.3. Decomposition of Spatial Differences

We characterised the spatial differences in the degree of coupling coordination between the digital economy and low-carbon development in the Yellow River Basin in 2011, 2014, 2017 and 2020 (Table 3).

**Table 3.** Results of the Decomposition of Spatial Differences in Coupling Coordination.

| Type of Differences | | 2011 | 2014 | 2017 | 2020 | Average Value |
|---|---|---|---|---|---|---|
| Overall Differences | | 0.0589 | 0.0614 | 0.0639 | 0.0703 | 0.0636 |
| Intra-regional Differences | Upper Reaches | 0.0481 | 0.0513 | 0.0485 | 0.0496 | 0.0494 |
| | Middle Reaches | 0.0523 | 0.0516 | 0.0530 | 0.0573 | 0.0536 |
| | Lower Reaches | 0.0516 | 0.0520 | 0.0535 | 0.0570 | 0.0535 |
| Inter-regional Differences | Middle—Upper | 0.0555 | 0.0557 | 0.0542 | 0.0604 | 0.0565 |
| | Middle—Lower | 0.0610 | 0.0673 | 0.0725 | 0.0783 | 0.0698 |
| | Upper—Lower | 0.0698 | 0.0724 | 0.0783 | 0.0892 | 0.0775 |
| Differences Contribution Rate | Intra-regional | 30.14% | 29.45% | 28.30% | 27.52% | 28.79% |
| | Inter-regional | 44.15% | 45.04% | 49.30% | 52.44% | 47.95% |
| | Super Variable Density | 25.70% | 25.52% | 22.41% | 20.03% | 23.26% |

1. Overall differences. The overall differences in coupling coordination over the study period widened, with the Gini coefficient rising from 0.0589 in 2011 to 0.0703 in 2020, an increase of 19.35%. With the transformation of China's economic development model, governments around the country have successively introduced supporting policies to support the construction of a digital economy and low-carbon development. Developed cities, with the advantages of capital, technology and markets, have accelerated the digitalisation and low-carbon synergistic transformation of their industries, gradually exacerbating the unbalanced development of coupling coordination for lagging cities.

2. Intra-regional differences. The Gini coefficients of the internal coupling coordination degree of the Yellow River in all regions of the study period increased. The mean values of the Gini coefficients of the coupling coordination in the upper, middle and lower reaches of the Yellow River were 0.0494, 0.0536 and 0.0535, respectively. The spatial differences in coupling coordination in the middle reaches of the Yellow River were the highest, mainly because the middle reaches of the Yellow River are located in the transition zone between high and low values of coupling coordination, whereas cities such as Xi'an and Taiyuan have particularly high and fast-growing levels of coupling coordination.

3. Inter-regional differences. The Gini coefficient between regions in terms of coupling coordination increased during the study period. The mean values of the Gini coefficient between the middle and upper reaches of the Yellow River, the middle and lower reaches, and the upper and lower reaches are 0.0565, 0.0698 and 0.0775, respectively. It can be observed that the great differences between the upper and lower reaches of the Yellow River became the main source of inter-regional differences.

4. Contribution to differences. The contribution of inter-regional differences in coupling coordination was greater than the contribution of intra-regional differences and hypervariable density during the study period, and the contribution of inter-regional differences increased (by 8.61%); the contribution of intra-regional differences and hypervariable density decreased.

### 5.1.4. Spatial Correlation Characteristics

We calculated the global Moran's I for the coupling coordination of the digital economy and low-carbon development from 2011 to 2020 and analysed the overall spatial correlation characteristics of the coupling coordination (Table 4). As shown in Table 4, all Moran's I were significantly positive ($Z > 2.58$, $p < 0.01$) during the study period, indicating that there is a positive spatial correlation between the coupling coordination degrees of 78 cities in the Yellow River Basin, i.e., cities with higher coupling coordination degrees tend to cluster, while cities with lower coupling coordination degrees also tend to cluster. Moran's I decreased, indicating that the spatial spillover effect of the coupling coordination degree shows a weakening trend, probably due to the different economic bases, resource factors and ecological environment conditions of the cities.

**Table 4.** Global Moran's I of Coupling Coordination in the Yellow River Basin.

| Year | I | Z | P | Year | I | Z | P |
|------|-------|-------|-------|------|-------|-------|-------|
| 2011 | 0.277 | 3.809 | 0.000 | 2016 | 0.277 | 3.811 | 0.000 |
| 2012 | 0.269 | 3.696 | 0.000 | 2017 | 0.271 | 3.730 | 0.000 |
| 2013 | 0.278 | 3.814 | 0.000 | 2018 | 0.268 | 3.704 | 0.000 |
| 2014 | 0.265 | 3.648 | 0.000 | 2019 | 0.263 | 3.642 | 0.000 |
| 2015 | 0.265 | 3.658 | 0.000 | 2020 | 0.266 | 3.685 | 0.000 |

We analysed the local autocorrelation of coupling coordination with LISA clustering plots for 2011, 2014, 2017 and 2020 (Figure 3). In 2011, the H-H clustering area contained 12 cities, including Qingdao, Jinan and Yantai, which have developed economies and convenient transportation. This provides sufficient guarantees for the coupling coordination of the digital economy and low-carbon development, as well as generating knowledge and technology spillover effects. This is a demonstration of the surrounding cities, resulting in a high level of overall regional coupling coordination. The L-H clustering area contains Dezhou, which, due to its natural conditions, resource status and other factors, has a low level of coupling coordination compared to the surrounding cities. The L-L clustering area contains eight cities, including Baiyin, Guyuan and Zhongwei, all of which have a low level of coupling coordination, and each city needs to find its strengths to break through established limitations. The H-L clustering area contains one city, Lanzhou. The clustering of digital industries has led to a greater digital divide between cities, and through the siphoning effect some cities acquire resource elements, creating negative externalities for neighbouring cities. The other cities did not pass the significance test, indicating that these cities are less connected to their neighbouring cities. From 2011 to 2020, the number of cities in the H-H clustering area increased slightly, and the regional scope extended from east to west. The number of cities in the L-L clustering area decreased, and the regional centre of gravity moved from south to north. The number of cities in the H-L clustering area increased, mainly in the upper and middle provincial capitals such as Taiyuan and Yinchuan. The number of cities in the L-L clustering area remained unchanged, and the regional scope changed from Jining to Rizhao. In sum, few cities experienced a jump in agglomeration type during the study period, and the local spatial agglomeration characteristics were generally stable. Over time, an H-H clustering area centred on Shandong and an L-L clustering area centred on south-eastern Gansu formed.

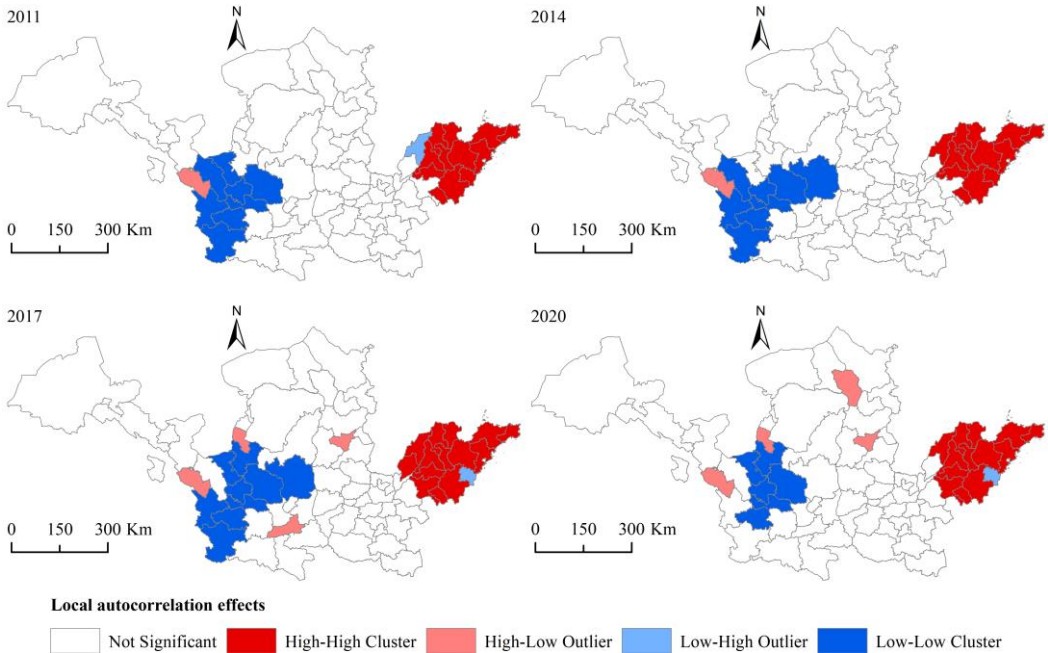

**Figure 3.** LISA Clustering Map of the Coupling Coordination in the Yellow River Basin.

### 5.2. Driving Factors of Coupling Coordination

The drivers were discretised by the natural breakpoint method and brought into the geodetector to obtain the q-values for the drivers in 2011, 2014, 2017 and 2020 (Table 5). Overall, the mean values of the driving forces for each driver during the study period are ranked as internet penetration rate (0.7770) > size of telecommunications industry (0.7448) > level of urbanisation (0.4685) > development of digital finance (0.4494) > level of human capital (0.4372) > upgrading of industrial structure (0.2419) > ecological environment support (0.1204) > government regulatory capacity (0.1130) > degree of market development (0.0611). All drivers contribute positively to the coupling coordination of the digital economy and low-carbon development in the Yellow River Basin, and hypotheses 1–9 are tested. The average driving force of factors within the digital economy system > the average driving force of factors within the low-carbon development system > the average driving force of external factors of both systems. Although the influence of each driver varies somewhat over time, internet penetration and the size of the telecommunications industry were consistently the most important factors driving the coupling coordinated development of the two systems.

**Table 5.** Single Factor Detection Results.

| Driving Factors | 2011 | 2014 | 2017 | 2020 | Average Value |
|:---:|:---:|:---:|:---:|:---:|:---:|
| X1 | 0.7320 | 0.7330 | 0.7644 | 0.8787 | 0.7770 |
| X2 | 0.2844 | 0.3511 | 0.5739 | 0.5882 | 0.4494 |
| X3 | 0.7334 | 0.7718 | 0.6994 | 0.7745 | 0.7448 |
| X4 | 0.3460 | 0.2840 | 0.0096 | 0.2539 | 0.2419 |
| X5 | 0.5277 | 0.4739 | 0.4736 | 0.3988 | 0.4685 |
| X6 | 0.1455 | 0.2125 | 0.0907 | 0.0328 | 0.1204 |
| X7 | 0.0568 | 0.0668 | 0.0895 | 0.2390 | 0.1130 |
| X8 | 0.0436 | 0.0328 | 0.0457 | 0.1223 | 0.0611 |
| X9 | 0.4196 | 0.4709 | 0.3469 | 0.5112 | 0.4372 |

Note: The above results are all significant at the 1% level.

#### 5.2.1. Internal Driving Factors

(1) Internet penetration rate. The drivers of internet penetration increased. The mean value of this driving force ranks first, indicating that the regions in the Yellow River Basin achieved results in areas such as network infrastructure construction and internet penetration and application. The information network has not only laid a solid foundation for the development of the digital economy but also made contributions to the low-carbon transformation of the economy and society. (2) Digital finance development. The drivers of digital finance development continue to grow and the mean value of the drivers is high. Digital finance has become a driving force for the coordinated development of the digital economy and low-carbon development across the basin. (3) Size of the telecommunications industry. The driver of the telecommunications industry scale increased, and the average value of this driver ranks second. Many telecommunications companies in the Yellow River Basin are making every effort to promote the green and low-carbon transformation of the telecommunications industry, the level of coupling coordination between the digital economy and low-carbon development has been further enhanced. (4) Industrial structure upgrading. The drivers of industrial structure upgrading increased, but the average value of the driving force is small. The Yellow River Basin is dominated by resource-based industries and upgrading the industrial structure is relatively slow, so the positive effect on the coupling coordinated development of the two systems remains small. (5) Level of urbanisation. As urbanisation in the Yellow River Basin is increasing rapidly, urban construction has led to a sharp increase in energy consumption and an increase in total carbon emissions, making the contribution of urbanisation to the coupling coordination of the digital economy and low-carbon development decline. (6) Ecological Environmental support. The drivers of ecology decrease, and the average value is only 0.1204. This is due to the fragile ecological environment of the Yellow River basin, overgrazing of pastures

and excessive deforestation, resulting in a decrease in carbon sink capacity. The application rate of digital technology in environmental protection and restoration is relatively low.

### 5.2.2. External Driving Factors

(1) Government regulatory capacity. The current drive for government regulatory capacity remains weak, and local governments in the Yellow River Basin should further leverage their financial resources to guide the process. (2) Degree in market development. The degree of market development is a driver of volatility. However, as the level of marketability is still low in most of the Yellow River Basin, the positive effects of the market on the coupling coordinated development of the two systems have not been fully released. (3) Human capital levels. Human capital level drivers increased, and the average value of the drivers is high. In recent years, projects such as the Yellow River Talent Programme have been used to train a large number of talented people in the digital and low-carbon economies, injecting new strength into the coordination and progress of the two systems.

### 5.2.3. Interaction Factor Detection

The drivers were averaged for the years 2011, 2014, 2017 and 2020 to obtain the interaction relationships for each driver (Table 6). Overall, all drivers interacted in an enhanced manner, with 18 sets of variables producing a non-linear enhancement effect. Specifically, the interaction of government regulatory capacity and market development with other variables produced a non-linear enhancement effect, indicating that government regulatory capacity and market development play a fundamental role in the coupling coordinated development of the two systems. Although the driving force in a single-factor detection of ecological support was weak, there was a strong driving force when factors acted together, e.g., two groups of variables interacting separately with ecological support to produce a non-linear enhancement effect. The remaining 27 groups of interaction terms show a two-factor enhancement effect. The coupling coordination of the digital economy and low-carbon development is the result of a combination of factors, with the interaction between all variables increasing.

**Table 6.** Interaction Factor Detection Results.

| Driving Factors | X1 | X2 | X3 | X4 | X5 | X6 | X7 | X8 | X9 |
|---|---|---|---|---|---|---|---|---|---|
| X1 | 0.8235 | | | | | | | | |
| X2 | 0.9506 | 0.6182 | | | | | | | |
| X3 | 0.8425 | 0.9119 | 0.6945 | | | | | | |
| X4 | 0.9538 | 0.7057 | 0.9200 | 0.2490 | | | | | |
| X5 | 0.9155 | 0.7004 | 0.8747 | 0.7208 | 0.5144 | | | | |
| X6 | 0.9345 | 0.7636 | **0.9220** | **0.4755** | 0.6232 | 0.1640 | | | |
| X7 | **0.9316** | **0.8608** | 0.8422 | 0.6300 | 0.8289 | 0.5994 | 0.0885 | | |
| X8 | 0.8563 | 0.7643 | 0.7875 | 0.5288 | 0.7354 | 0.5029 | 0.2709 | 0.0322 | |
| X9 | 0.8628 | 0.7990 | 0.8380 | **0.7213** | 0.7029 | 0.5518 | **0.7159** | **0.6371** | 0.4225 |

Note: bold values indicate non-linear enhancement effects.

## 6. Conclusions and Recommendations

### 6.1. Conclusions

Based on panel data from 78 cities in the Yellow River Basin from 2011 to 2020, this paper explores the spatial-temporal characteristics and drivers of the degree of coupling coordination between the digital economy and low-carbon development and draws the following conclusions.

(1) The digital economy index in the Yellow River Basin grew rapidly over the study period, whereas the low-carbon development index grew slowly, and digital economy development lagged behind low-carbon development. The coupling coordination of the two systems grew steadily and moved from the near-disorder stage to the primary coordination stage. Among them, 2011–2012 was in the near-disorder stage, 2013–2019 was in the basic

coordination stage and 2020 was in the primary coordination stage. In addition, the overall level of coupling coordination was still low and needs to be further improved.

(2) The spatial non-equilibrium characteristics of the coupling coordination degree of the two systems were significant, showing a spatial distribution characteristic of lower reaches > middle reaches > upper reaches, and the coupling coordination level was higher in provincial capitals and coastal cities such as Xi'an, Jinan, Zhengzhou and Qingdao, whereas in most other cities it was relatively low. In addition, the spatial differences in the coupling coordination degree increased during the study period, and the differences between regions, especially between the upper and lower reaches of the Yellow River, were the main source of the overall differences.

(3) There was a significant positive spatial correlation between the degree of coupling coordination of the two systems, with areas of higher coupling coordination tending to cluster, and areas of lower coupling coordination tending to cluster as well, but the correlation is weaker. The local spatial clustering characteristics were dominated by the H-H clustering area in Shandong and the L-L clustering area in south-eastern Gansu, and the local spatial clustering characteristics were generally stable. The linkage effect between cities and the driving effect of the high-value areas in the middle and upper reaches needs to be strengthened.

(4) The degree of coupling coordination between the digital economy and low-carbon development in the Yellow River Basin is influenced by a combination of internal and external factors. The average driver of internal factors for the digital economy system > the average driver of internal factors for the low-carbon development system > the average driver of external factors for both systems, with internet penetration and the size of the telecommunications industry always the dominant drivers. In addition, the interactions between the different drivers were all enhanced.

### 6.2. Recommendations

Synthesising the above findings, suggestions for optimising the coupling coordinated relationship between the digital economy and low-carbon development in the Yellow River Basin are proposed.

(1) Promote the construction of the digital economy in all aspects. The lagging development of the digital economy in the Yellow River Basin is a critical problem, and the comprehensive development of the digital economy should be improved in three dimensions: digital infrastructure, digitalisation of industry and digital industrialisation. First, it is necessary to coordinate the planning of the regional layout of digital infrastructure, increase the investment in information networks, and guide the direction of investment in information networks. Second, strengthen the research and development of core technologies; actively explore digital technology application scenarios; promote digital transformation in agriculture, industry and service sectors; and deepen the application of digital technology in production and operations. Third, accelerate the cultivation and growth of core industries of the digital economy, such as mobile communications and software manufacturing, and enhance their ability to serve economic and social development.

(2) Enhance the coordinated interaction between the digital economy and low-carbon development. At present, the level of coupling coordination between the digital economy and low-carbon development in most cities in the Yellow River Basin is still low, and the interaction between the two systems should be further enhanced. First, the digital economy should be stimulated and its structural optimisation and resource allocation effects should be employed, thereby promoting the transformation of traditional energy-intensive industries. The scope of application of digital technology in new energy development and ecological protection should be expanded to enhance energy efficiency and carbon sink capacity. Second, with the goal of "Carbon Neutrality" as a guide, implement the concept of low-carbon development, accelerate the low-carbon transformation of high energy-consuming areas such as 5G infrastructure and big data centres, and continue to optimise the development model of the digital economy. High-quality development of new

urbanisation should be accelerated to further expand consumer demand for digital products or services. Third, facilitate the government's ability to regulate, control and increase financial support to strengthen comprehensive coordination. Market vitality should be stimulated, information technology platforms built that are oriented towards green and low-carbon development and digital technology needs, and competitive market environments cultivated for virtuous circular development. The ability of society to participate should be expanded, and the cultivation of talent facilitated with a combination of digital and low-carbon expertise, injecting new momentum for the coordinated growth of the digital economy and low-carbon development.

(3) Steady progress towards coordinated regional development. The spatial differences in the degree of coupling coordination between the digital economy and low-carbon development in the Yellow River Basin are obvious, and the differences between cities are increasing. According to the economic development and resource and environmental conditions of each region, an accurate assessment of competitive advantages should be made and policies tailored to local conditions. The lower reaches of the Yellow River are well-funded and talented, and should actively undertake the task of tackling the core digital technologies. On the one hand, it is necessary to strengthen technical exchanges between the government, universities and enterprises, and establish a regular and long-term cooperation mechanism. On the other hand, international cooperation and exchange should be strengthened to build a world-class digital industry cluster. Meanwhile, sound laws and regulations on market access, operation and supervision in the field of low-carbon development create a strong institutional guarantee for the low-carbon transformation of enterprises. The middle reaches of the Yellow River are still on the rise in terms of digital economy development and have a high proportion of high-carbon industries. The awareness and ability of enterprises in the middle reaches of the river should be enhanced. Furthermore, the digital transformation of industrial chains should be accelerated by creating digital demonstration platforms and fostering leading enterprises to enhance the empowering effect of the digital economy on high-energy-consuming industries. Using metropolitan areas and city clusters as the main carriers, the agglomeration and influence of regional digital and low-carbon industries should be enhanced, thereby improving the driving power of provincial capitals such as Xi'an and Taiyuan on neighbouring cities. The upper reaches of the Yellow River can consider building a mega data processing centre based on the low-cost advantage and resource endowment conditions, to drive up the level of digital infrastructure, while accelerating the development of the carbon sink industry and promoting the organic integration of the digital and carbon sink industries. To further narrow regional differences, the leading role of the lower Yellow River to the middle and upper reaches should be strengthened, and the "Belt and Road" strategy should be used to guide the convergence of financial capital, markets and talents from the lower reaches to the middle and upper reaches, and enhancing the coupling coordination of the overall digital economy and low-carbon development in the Yellow River basin by expanding the technology and knowledge spillover effect.

### 6.3. Limitations and Future Research

Limitations of this paper: Firstly, although the indicator system is constructed based on the connotation of the digital economy and low-carbon development, it is still not comprehensive enough due to the availability of data. Secondly, the scope of the drivers of the coupling coordination degree of the digital economy and low-carbon development needs to be further expanded. Thirdly, this paper has only conducted research at the prefecture-level city, and has not studied the coupling coordination relationship between the digital economy and low-carbon development at the county level. Therefore, in the future, we will further enhance the research capability of the team to supplement the comprehensive evaluation index system of the digital economy and low-carbon development as well as the drivers of the coupling coordinated development of the two systems, and further analyse

the coupling coordinated development relationship between the two systems from the county level.

**Author Contributions:** Conceptualisation, Z.X.; methodology, Z.X.; software, Z.X.; Writing—original draft, Z.X.; formal analysis, F.C.; supervision, F.C. All authors have read and agreed to the published version of the manuscript.

**Funding:** This research was funded by General Projects of the National Social Science Foundation of China, grant number 17BJY041 and Shandong Provincial Social Science Planning Research Key Project, grant number 20BJJJ06 and Shandong Provincial Social Science Planning Research Project, grant number 21DGLJ23.

**Institutional Review Board Statement:** Not applicable.

**Informed Consent Statement:** Not applicable.

**Data Availability Statement:** No applicable.

**Conflicts of Interest:** The authors declare no conflict of interest.

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
