# Peer review of "Spatial-Temporal Characteristics and Driving Factors of Coupling Coordination between the Digital Economy and Low-Carbon Development in the Yellow River Basin"

_sustainability, doi:10.3390/su15032731_

Round 1
Reviewer 1 Report
The issues raised by the authors are topical and interesting for the reader. It can be a source of inspiration for both theorists and practitioners. The entire text is comprehensive and clear and the scientific argument is logical, but nevertheless the text needs improvement.
The title indicates the topic covered in the text.
The abstract is well written, comprehensive and concise enough, but it should be supplemented with the purpose, the other elements are included. The keywords are appropriate.
The introduction also includes a literature review, which should be described as a separate section. In the introduction, it should be clearly indicated what is the purpose of the text and its structure.
The authors critically reviewed the literature. It is moderately extensive but adequate to the subject matter and up-to-date.
In the methodological section, the authors should identify the research problems and research hypotheses, and refer to these elements in the next section.
The analyses are based on secondary data, however, the authors should clearly indicate the limitations of the research (shortcomings) and the directions for further research. There is no separate discussion.
Summarizing, the text is interesting, but it needs some corrections and additions.
Reviewer 2 Report
This manuscript explores the spatial-temporal characteristics and drivers of the degree of coupling coordination between the digital economy and low-carbon development.The relevant research methods are relatively complete, and the conclusions are clear and instructive. The reviewer recommend that the journal accept the manuscript and publish it.
Author Response
Dear Reviewers.
Thanks for your approval of our articles, we'll keep up the good work!
Reviewer 3 Report
Dear authors,
Overall, I liked this paper a lot. It is well-written, well-structured and it touches upon some of the contemporary issues regarding sustainability and environmental development.
I propose some minor revisions as below:
1. At the Introduction, expand a bit please on when the digital transformation of China began.
2. Likewise, elaborate please around the carbon emission providing some facts and statistics if possible.
3. As you say the Yellow River Basin, is a geo-economic area. Please elaborate further on which specific areas or countries are directly or indirectly affected.
4. The methodological effort is very ambitious, therefore please provide some examples of coupling at section titled 'Coupling Mechanism'. Without explicating how coupling can be applied, your methodological justification might seem week or non-supported.
5. Most importantly at the conclusion and recommendations sections, please elaborate with more geographical specific and tangible results, where these applications (digital) can be achieved. How can the middle upper class cities aid other small cities? What do you mean by synergistic innovation? You sound very ambitious but do you identify any obstacles and limitations? Beraucracy or lack of speed in the implementation of these infrastactures?
I hope you will find these comments helpful,
Reviewer 4 Report
In this manuscript, the level of coupling coordination between the digital economy and low-carbon development in the Yellow River Basin is measured, and the spatial-temporal characteristics and driving factors are analyzed. The content has some meanings, but there are also some shortcomings.
Scientific research is not playing with concepts, not creating concepts. The author should introduce what is the digital economy at first? Many of the indicators used do not reflect the so-called "digital economy" and "low-carbon development", Such as Mobile phone users, Urban registered unemployment rate, Urbanization level, etc.
In the part of 4.2.1Driving Factor, Some factors may be driving factors, while some may not be driving factors and only can be as the influence factors.
Table5. The meanings of the letter(X1,X2,X3,,,,,) need to interpretation.
Round 2
Reviewer 4 Report
The quality of the manuscript has been improved, so I agree to accept the revised article.